# Determination of X-ray detection limit and applications in perovskite X-ray detectors

Lei Pan [1], Shreetu Shrestha[2], Neil Taylor[1], Wanyi Nie [2] & Lei R. Cao [1✉]

X-ray detection limit and sensitivity are important figure of merits for perovskite X-ray detectors, but literatures lack a valid mathematic expression for determining the lower limit of detection for a perovskite X-ray detector. In this work, we present a thorough analysis and new method for X-ray detection limit determination based on a statistical model that correlates the dark current and the X-ray induced photocurrent with the detection limit. The detection limit can be calculated through the measurement of dark current and sensitivity with an easy-to-follow practice. Alternatively, the detection limit may also be obtained by the measurement of dark current and photocurrent when repeatedly lowering the X-ray dose rate. While the material quality is critical, we show that the device architecture and working mode also have a significant influence on the sensitivity and the detection limit. Our work establishes a fair comparison metrics for material and detector development.

[1] Nuclear Engineering Program, Department of Mechanical and Aerospace Engineering, The Ohio State University, Columbus, OH, USA. [2] Center for Integrated Nanotechnology Materials Physics and Application Division, Los Alamos National Laboratory, Los Alamos, NM, USA. ✉email: cao.152@osu.edu

X-ray detectors see wide applications in security inspection, academic research, industry, and medical imaging[1–4]. Metal halide perovskites emerged recently as promising candidates for next generation direct conversion X-ray detectors[5,6]. Research efforts on perovskite X-ray detectors have seen surged publications in recent years. The high quality perovskite single crystals can be synthesized from a low-cost solution grown method with excellent performance for X-ray and gamma-ray detection[7–11], and perovskite thin films can be flexibly deposited or printed onto various substrates in a large area for X-ray detection[12–15].

The detection limit of X-ray dose rate (alternatively, lower limit of detection or the lowest detectable dose rate) and sensitivity are important figure of merits for X-ray detectors. However, the methodology on how to properly determine the detection limit has not been able to catch up with the rapid development of perovskite materials and their applications for X-ray detection. Many publications[12–14,16–19] have only reported the sensitivity of their perovskite X-ray detectors, while lacking the measure of the detection limits (Fig. 1). Among those that have reported the detection limit (Fig. 1), some papers[7,9,20] did not present clearly the method for their detection limit determination, and some[21–25] claimed the use of definitions based on the 1975 International Union of Pure and Applied Chemistry (IUPAC) detection limit definition[26] without an explicit equation provided. Although some of the work[27–29] have also adopted a method referenced as derivatives of the IUPAC definitions, there has been no discussions on its statistical validity. The ambiguity and misleading concepts on X-ray detection limit in the current literatures call for the urgent needs to establish a method that is accurate and easy to use.

In this work, we propose a systematic approach for the detection limit determination of an X-ray detector based on a well-established statistical model that correlates the dark current and photocurrent under X-ray irradiation quantitatively with the detection limit. Through a review and comparison of the currently practiced methods[27–29], the original IUPAC definitions[26], and the well-known Currie method[31] where the IUPAC definitions stems from, we show that our method is statistically strict and accurate. Specifically, detection limit of X-ray dose rate $\dot{D}_{limit}$ can be obtained by calculation from a measurement of the device's dark current $I_{dark}$ and the sensitivity $s$ as a calibration factor, which is, for simplicity, named as dark current ($I_{dark}$ & $s$) method thereafter. Alternatively, $\dot{D}_{limit}$ may also be obtained by the repetitive measurements of $I_{dark}$ and detector photocurrent under X-ray irradiation $I_{X-ray}$ when repeatedly lowering the X-ray

dose rate, which is named as X-ray photocurrent ($I_{dark}$ & $I_{X-ray}$) method. Despite the two approaches, we recognize that the $I_{dark}$ & $I_{X-ray}$ method requires a laborious experimental work by decreasing the X-ray dose rate successively to approach the background, and it yields asymptotic $\dot{D}_{limit}$ with a large uncertainty depending on the experiment instrumentation and human factors, e.g., the X-ray dose rate used in experiment. Contrarily, the $I_{dark}$ & $s$ method yields $\dot{D}_{limit}$ independent of X-ray dose rate, which represents the intrinsic detector performance of measuring a small dose rate.

In this work, we also compare the sensitivity of perovskite X-ray detectors made of methylammonium lead triiodide (MAPbI$_3$) single crystal with different device architecture and different operation mode, e.g., photodiode in reverse bias vs forward bias mode, since device sensitivity $s$ is needed as a calibration factor for the $I_{dark}$ & $s$ method. We conclude that the device architecture and operation mode have a significant influence on the value of sensitivity $s$ and detection limit $\dot{D}_{limit}$ to be measured, but our proposed methodology itself for the determination of detection limit is independent of the material's properties and device's operation, which could be used as an evaluation standard for materials quality and detector performance comparison.

## Results and discussions

**A statistical model based method for detection limit determination.** The statistical foundation used in the practiced method[27–29] for detection limit determination traces back to the 1975 IUPAC definitions of detection limit and further back to the method proposed by Currie in 1968[31]. In this work, we apply a statistical model to establish methods in determining X-ray detection limit that extends the well-known Currie formulars that have been widely cited and universally incorporated into many international standards and regulations[32–34]. Currie method stems from Bernoulli process where atoms are counted over time based on their radioactive decay and assumes Normal distribution for the blank signal $X_B$, defined as the signal resulting from measurement where the substance sought is absent, and for the gross signal $X$ where the substance sought may exist. Three key parameters related to the detection limit are established, that is, the critical level $L_C$, the qualitative detection limit $L_D$, and the determination limit for quantification $L_Q$. If the measured gross signal $X$ is higher than $X_c$ that is equal to the sum of the mean of blank signal $\bar{X}_B$ and the critical level $L_C$, i.e., $X > X_c, X_c = \bar{X}_B + L_C$, a binary decision is made, i.e., "detected" is reported with a false positive probability of $\alpha$ (type I error),

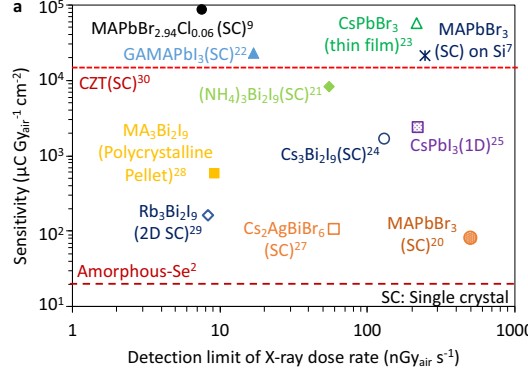

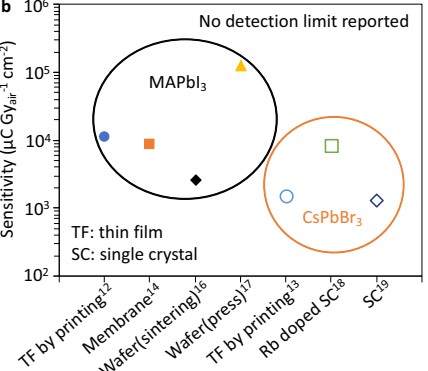

**Fig. 1 Detection limit and sensitivity of various perovskite X-ray detectors. a** Perovskite X-ray detectors with sensitivity and detection limit reported in the literatures with comparison to amorphous Selenium[2] and Cadmium Zinc Telluride (CZT)[30]. **b** Sensitivity of MAPbI$_3$ and all-inorganic CsPbBr$_3$ X-ray detectors prepared with different material synthesizing methods and device structures where no detection limits are reported. The superscripts stand for the reference number.

**Table 1 $L_C$, $L_D$, and $L_Q$ in the Currie method[31].**

| | $L_C$ | $L_D$ | $L_Q$ |
|---|---|---|---|
| Paired observations | 2.33 $\sigma_B$ | 4.65 $\sigma_B$ | 14.1 $\sigma_B$ |
| "Well-known" blank | 1.64 $\sigma_B$ | 3.29 $\sigma_B$ | 10 $\sigma_B$ |

Paired observation is defined by Currie as equivalent observations of sample (plus blank) and blank. "Well-known" blank is defined by Currie as a long history of observations of the blank. $L_C$, critical level; $L_D$, qualitative detection limit; $L_Q$, determination limit for quantification; $\sigma_B$, blank signal's standard deviation.

otherwise "not detected" is reported with a false negative probability of $\beta$ (type II error). When both $\alpha$ and $\beta$ are set to be 5%, Currie provided a set of convenient working formulars to calculate $L_C$, $L_D$, and $L_Q$ for a large majority of radiation counting applications, which is shown in the widely reprinted Table 1[31]. Several assumptions are made for Table 1: (1) the standard deviation of the blank signal and the gross signal, denoted as $\sigma_B$ and $\sigma_Q$, respectively, are approximately constant and equal to each other in the considered range; (2) the relative standard deviation of the signal, that is $\sigma_Q/L_Q$, is set to be 10% for the determination limit $L_Q$.

Following the Currie method, a smallest detectable gross signal $X_D$ can be directly calculated as the mean of blank signal $\bar{X}_B$ plus $L_D$, i.e., $X_D = \bar{X}_B + L_D$. Similarly, a smallest quantifiable gross signal $X_Q$ is calculated as $X_Q = \bar{X}_B + L_Q$. To estimate $\bar{X}_B$ and $\sigma_B$, a measurement of the blank signal is necessary. Although the underlying assumption of Table 1 is Normal distribution in Currie's classical paper[31], the radioactive decay problems being delt with by Currie's method originate from Bernoulli process and end up with Poisson-Normal distribution, defined as the Normal distribution $\bar{X}_B \sim N(\mu, \sqrt{\mu/n})$ transitioned from Poisson distribution $\bar{X}_B \sim (1/n)\text{Poisson}(n\mu)$ when the mean $\mu$ of Poisson distribution is large, where n stands for the sample size that is a statistical terminology defined as the number of individual samples measured in an experiment. In γ-ray photon counting problems, the sample size n refers to the number of repetitive time-accumulated counting events the experimenter has made. Each sample counting is lasting for a certain amount of time and have different counts accumulated.

On the implementation level, the original Currie's definition of the Paired observations and the "Well-known" blank (Table 1) does not explicitly distinguish the sample size n. We interpret the Paired observations being the case that has only one counting ($n = 1$) of the blank signal for $\bar{X}_B$ and $\sigma_B$ estimation, and the "Well-known" blank being the case with many counting events ($n \gg 1$) acquired of the blank signal. With the Poisson-Normal distribution assumption, the $\sigma_B$ can be estimated as $\sqrt{\bar{X}_B}$ when only one long time counting ($n = 1$) event for the Paired observations case is acquired. Both Paired observations and "Well-known" blank cases assume implicitly that only one counting event ($n = 1$) of the gross signal would be acquired.

Adopting the idea of the Currie method, the IUPAC definition sets the smallest detectable gross signal $X = X_C$ that is higher than the mean of the blank signal $\bar{X}_B$ by a difference of $3\sigma_B$[26], i.e., $X_C = \bar{X}_B + 3\sigma_B$, which is equivalent to setting $L_C = 3\sigma_B$. Consequently, the IUPAC defined detection limit ensures a type I error $\alpha = 0.13\%$, but a type II error $\beta = 50\%$[32,35]. The IUPAC definition also assumes implicitly only one sample ($n = 1$) for the gross signal would be acquired as it is descended from the Currie method. The relationship of the relevant parameters for the "Well-known" blank in Currie method and the IUPAC definition are illustrated in Fig. 2a.

Although the practiced method[27–29] for X-ray detection limit determination is referenced on the IUPAC definition of detection limit, we show there is a discrepancy between IUPAC definition and the practiced method[27–29]. In X-ray detector testing, the blank signal is the dark current $I_{dark}$ and the gross signal is the photocurrent under X-ray irradiation $I_{X-ray}$. The net current $I_{net}$ is the difference between the photocurrent under X-ray $I_{X-ray}$ and the dark current $I_{dark}$, i.e., $I_{net} = |I_{X-ray} - I_{dark}|$, as shown in Fig. 2b. In the IUPAC definition, the noise value is taken as the standard deviation of the blank signal, $\sigma_B$, as it follows the Currie method assuming the standard deviation is the same for blank signal and for gross signal, that is, $\sigma_B = \sigma_Q$. However, the noise current $I_{noise}$ taken as the standard deviation of photocurrent under X-ray $\sigma_{I_{X-ray}}$ will typically be larger than the noise current taken as the standard deviation of dark current $\sigma_{I_{dark}}$ due to the fluctuation of X-ray photon flux, X-ray generated charge carrier generation and recombination. The practiced method[27–29], shown in Fig. 2b, considered such difference by simply replacing $\sigma_{I_{dark}}$ in the IUPAC definition with $\sigma_{I_{X-ray}}$, which lacks statistical validity and results in an improper equation to use. Besides, the original IUPAC definition assumes only one sample of the gross signal will be measured, but in reality, the sample number (i.e., number of digitally sampled current data points) of the $I_{X-ray}$ may be far larger than 1.

A simple modification to the IUPAC definition would not result in a correct equation to use for detection limit determination because of the different physics involved. IUPAC definition follows the Currie method that is applicable for γ-ray photon counting where the physics behind is radioactive decay. However, in X-ray detection, the large quantity of X-ray photons emitted from X-ray machine is not a decay phenomenon in nature, nor does the dark current $I_{dark}$ and the photocurrent under X-ray $I_{X-ray}$ of an X-ray detector. The statistical distribution applicable to photon counting is Poisson-Normal distribution where the sample size n that refers to number of counting events performed, is typically 1 or not far larger than 1. Contrarily, $I_{dark}$ and $I_{X-ray}$ for an X-ray detector follows Normal distribution, where the sample size n in the electric current measurement referred as the number of digitized current points is typically very large.

To establish a proper procedure for the detection limit determination of an X-ray detector working in current mode, we propose a method by comparing the means of two normally distributed samples (i.e., physical parameters) of unequal sample size (i.e., number of sampled data points), which considers the possible different standard deviations of the dark current $I_{dark}$, the photocurrent under X-ray $I_{X-ray}$, and the large number of digitized current data points. The model containing Eqs. (1) and (2) (named as Detection Limit (DL) equations thereafter in this paper) are a well-established model in statistical hypothesis testing theory[36]. In DL equations, $n_1$, $n_2$ are the sample size of the Group 1 and Group 2, respectively. $\Delta = |\mu_1 - \mu_2|$ is the theoretical detection limit. $(\mu_1, \sigma_1^2)$, $(\mu_2, \sigma_2^2)$ are the means and variances of the two respective groups following Normal distribution. $k = n_2/n_1$ is the ratio of the two sample sizes. The $z_{1-\alpha}$ and $z_{1-\beta}$ are the Z-score corresponding to type I error $\alpha$ and type II error $\beta$, respectively, for one-sided test. The relevant parameters are illustrated in Fig. 2c. The Eqs. (1) and (2) can be rearranged to cancel k, yielding Eq. (3), which is another form of the DL equations.

$$n_1 = \frac{(\sigma_1^2 + \sigma_2^2/k)(z_{1-\alpha} + z_{1-\beta})^2}{\Delta^2} \quad (1)$$

$$n_2 = \frac{(k\sigma_1^2 + \sigma_2^2)(z_{1-\alpha} + z_{1-\beta})^2}{\Delta^2} \quad (2)$$

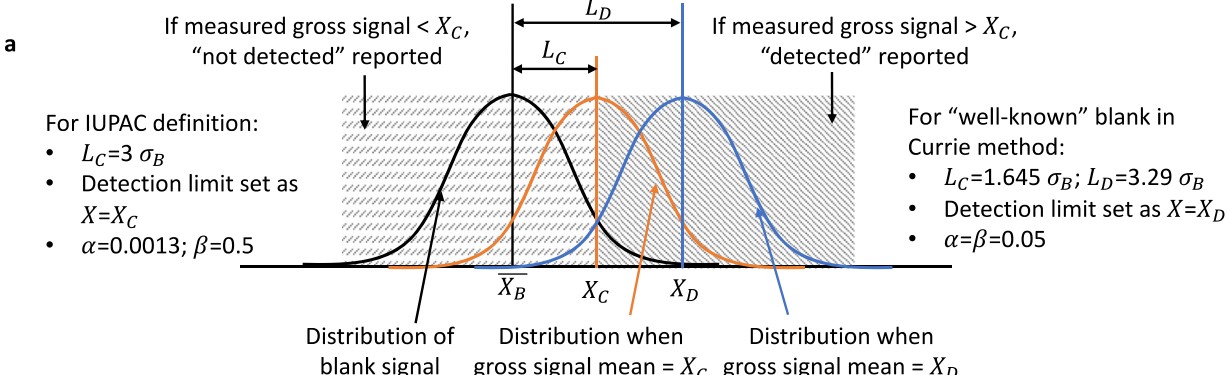

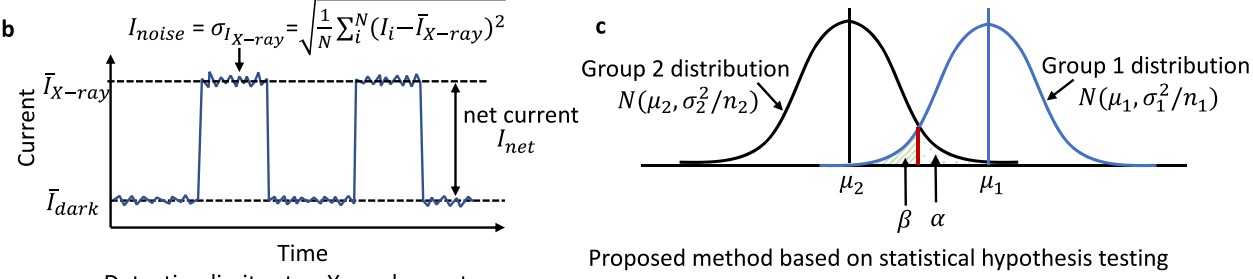

**Fig. 2 Review and comparison of different methods for detection limit determination. a** The "Well-known" blank in the Currie method and the IUPAC definition. $L_C$, critical level; $L_D$, detection limit; $\sigma_B$, blank signal standard deviation; $\alpha$ and $\beta$, type I and type II error; $X_D$, smallest detectable gross signal; $\bar{X}_B$, mean of the blank signal. **b** The practiced way[27–29] for perovskite detector dose rate detection limit determination. N is the total number of digitized electric current data points and $I_i$ is the i[th] point. **c** The method proposed in this work, which is reduced to the Currie method with certain pre-assumptions. ($\mu_1$, $n_1, \sigma_1^2$), ($\mu_2$, $n_2, \sigma_2^2$) are the means, sample sizes, and variances of the two respective groups following Normal distribution.

$$\Delta = \left|\mu_1 - \mu_2\right| = (z_{1-\alpha} + z_{1-\beta})\sqrt{\frac{\sigma_1^2}{n_1} + \frac{\sigma_2^2}{n_2}} \qquad (3)$$

Assuming Group 1 and Group 2 represent the gross signal (photocurrent under X-ray $I_{X-ray}$ in the case of X-ray detector) and the blank signal (dark current $I_{dark}$ in the case of X-ray detector) throughout in this work (equivalently vice versa), respectively, then a *prior* detection limit $\Delta = \left|\mu_1 - \mu_2\right|$ can be calculated with the measured blank signal of Group 2, i.e., ($\mu_2, \sigma_2^2$) and $n_2$ from the DL equations, with some necessary pre-assumptions (we use the term "*prior*" because this method requires pre-assumptions from a statistical perspective). We make assumptions consistent with the Currie method and show that the DL equations is same as Currie formulars when sample sizes (i.e., number of sampled data points) are being reduced (Fig. 2c). Instead of setting $\alpha = 0.13\%$ and $\beta = 50\%$ as in the IUPAC definition, we set more properly $\alpha = \beta = 0.05$, corresponding to $z_{1-\alpha} = z_{1-\beta} = 1.645$, which is consistent with Currie formulas. The standard deviation of the gross signal and the blank signal are assumed approximately equal as is in the Currie formulars, i.e., $\sigma_1^2 = \sigma_2^2$, when the gross signal is small approaching the level of detection limit. The influence of sample size (i.e., number of sampled data points) on the detection limit is reflected by the value of $k = n_2/n_1$. If we set $n_2 = n_1 = 1$, we have $k = 1$. Then the DL equations reduce to $\Delta^2 = (\sigma_1^2 + \sigma_2^2)(z_{1-\alpha} + z_{1-\beta})^2$. With the assumptions of $\sigma_1^2 = \sigma_2^2$ and $z_{1-\alpha} = z_{1-\beta} = 1.645$, we have $\Delta = \left|\mu_1 - \mu_2\right| = \sqrt{2}\sigma_2 * 3.29 = 4.65\sigma_2$, which is the detection

limit $L_D$ in Paired observations case in Currie's classical paper[31] (Table 1). If we set $n_1 = 1$ and $n_2 \gg n_1$, that is, $k = n_2/n_1 \gg 1$, so that $\sigma_1^2 \gg \sigma_2^2/k$ or $k\sigma_1^2 \gg \sigma_2^2$, then the DL equations reduce to $\Delta^2 = \sigma_1^2(z_{1-\alpha} + z_{1-\beta})^2$. With the same assumptions, we have $\Delta = \left|\mu_1 - \mu_2\right| = 3.29\sigma_2$, which is the detection limit $L_D$ in "Well-known" blank case in Currie's paper[31] (Table 1). The DL equations are based on Normal distribution, which makes it applicable to X-ray detectors working in a continuous current mode.

The DL equations indicate that the detection limit can be effectively reduced by increasing the sample size, i.e., $n_1$ or $n_2$. For example, in the case of X-ray detector, we may take the sample size of the dark current $n_2$ (i.e., $n_{I_{dark}}$, number of digitized points of dark current) to be a large number while the sample size of the photocurrent under X-ray $n_1$ (i.e., $n_{I_{X-ray}}$) to be a small number. Under such settings, if $n_2 \gg n_1$ and $n_1 = 1$, we can calculate a conservative detection limit of net current $I_{limit} = \Delta = \left|\mu_1 - \mu_2\right| = 3.29\sigma_2 = 3.29\sigma_{I_{dark}}$, equivalent to "Well-known" blank in Currie method. In theory but not in practice, if and only if $n_2 \gg n_1$ and $n_1 > 1$, the detection limit of net current $I_{limit}$ could be further reduced through calculation of $I_{limit} = \Delta = \sigma_2(z_{1-\alpha} + z_{1-\beta})/\sqrt{n_1} = 3.29\sigma_{I_{dark}}/\sqrt{n_{I_{X-ray}}}$. It is worth to note that the calculation so far only yields the detection limit of net current $I_{limit}$. The detection limit of X-ray dose rate $\dot{D}_{limit}$ requires $I_{limit}$ and sensitivity $s$ as calibration factor, which will be discussed in the later section.

In the dark current ($I_{dark}$ & $s$) method, no measurement of photocurrent under X-ray $I_{X-ray}$ is needed as our pre-assumptions

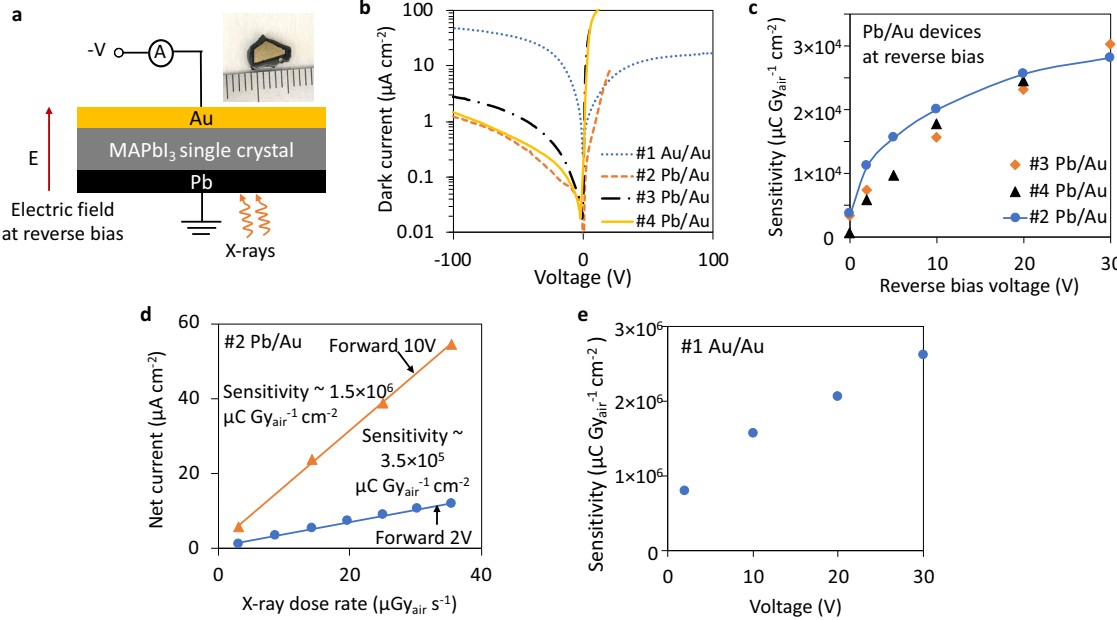

**Fig. 3 Sensitivity comparison of MAPbI$_3$ single crystal devices in different working mode. a** Experiment setup for reverse bias operation of the Pb/Au devices and the hole dominantly induced signal collection. **b** Dark *I–V* characterization of the devices. **c** Sensitivity of the reversely biased Pb/Au photodiode. **d** Sensitivity of the forward biased #2 Pb/Au photodiode. **e** Sensitivity of the Au/Au photoconductor.

set the standard deviation $\sigma_1$ of $I_{X-ray}$ equals to the dark current's standard deviation $\sigma_2$, and fix the values of $n_1$ and $n_2$. Practically, $I_{X-ray}$ can be measured to obtain its standard deviation $\sigma_1$ (i.e., $\sigma_{I_{X-ray}}$), sample size $n_1$ (i.e., $n_{I_{X-ray}}$), and mean value (i.e., $\mu_{I_{X-ray}}$). In combination with the measured dark current's standard deviation $\sigma_2$ (i.e., $\sigma_{I_{dark}}$), sample size $n_2$ (i.e., $n_{I_{dark}}$), and mean value (i.e., $\mu_{I_{dark}}$), we can perform *a posterior* check of the detectability (we use term "*posterior*" because this examination is performed after the measurement of $I_{X-ray}$ and $I_{dark}$). Specifically, with $n_1$, $n_2$, $\sigma_1^2$, $\sigma_2^2$ determined by measurement, the detection limit $\Delta = (z_{1-\alpha} + z_{1-\beta})\sqrt{\sigma_1^2/(n_1) + \sigma_2^2/(n_2)}$ can be calculated from the DL equations. Besides, a difference of the measured two mean values $|\mu_{I_{X-ray}} - \mu_{I_{dark}}|$ can also be obtained. We compare the calculated detection limit $\Delta$ with the measured mean value difference $|\mu_{I_{X-ray}} - \mu_{I_{dark}}|$. If $|\mu_{I_{X-ray}} - \mu_{I_{dark}}| > \Delta$, the photocurrent under X-ray $I_{X-ray}$ is detected with the preset false positive probability $\alpha$ and the false negative probability $\beta$ satisfied, otherwise the preset false positive and false negative probability cannot be satisfied. Following the principles above, we can successively lower the X-ray dose rate used to generate the $I_{X-ray}$ until $I_{X-ray}$ cannot be detected against the dark current. The lowest X-ray dose rate that generates the smallest detectable $I_{X-ray}$ is taken as the dose rate detection limit $\dot{D}_{limit}$. Although this method needs the measurement of $I_{dark}$ and $I_{X-ray}$, we simplify the name by calling it the X-ray photocurrent method ($I_{dark}$ & $I_{X-ray}$ method).

The experimental procedure of our $I_{dark}$ & $I_{X-ray}$ method is essentially the same as the practiced method[27–29], but we present proper statistical equations for which the method is based upon. Apparently, the experimental procedures of the $I_{dark}$ & $I_{X-ray}$ method require a repetitive and tedious work, and it can only produce an asymptotic value that will always be larger than the $\dot{D}_{limit}$ obtained from the dark current ($I_{dark}$ & $s$) method. Furthermore, the $I_{dark}$ & $I_{X-ray}$ method could yield a large uncertainty of the measured $\dot{D}_{limit}$ due to the limitation of the experiment instruments, e.g., X-ray tube dose rate range, number

of attenuators used. On the other hand, the dark current method only needs to measure dark current $I_{dark}$ and sensitivity $s$, which is easy and quick to perform and yields an $\dot{D}_{limit}$ independent of X-ray dose rate. Similar to the detection limit, the determination limit for quantification obtained by $I_{dark}$ & $I_{X-ray}$ or by $I_{dark}$ & $s$ method can be obtained by setting $z_{1-\alpha} + z_{1-\beta} = 10$ in the DL equations that is consistent with a relative standard deviation of 10% in Currie method.

**Sensitivity of MAPbI$_3$ detector in different working mode**. The measurement of the dark current can produce a calculated detection limit of net current $I_{limit}$. To convert $I_{limit}$ to the detection limit of X-ray dose rate $\dot{D}_{limit}$ (interchangeably, the lowest detectable dose rate), we need a calibration factor[31,35], which, in the case of perovskite X-ray detector, is the sensitivity $s$. The relationship between $I_{limit}$ and $\dot{D}_{limit}$ is $\dot{D}_{limit} = I_{limit}/(A * s)$, where $A$ is the readily known detector area and sensitivity $s$ should be measured under the same bias voltage condition as the dark current measurement.

Material quality is critical in determining the device X-ray sensitivity, but the device architecture and operation mode also have a significant influence on the sensitivity. To demonstrate such influence, we tested four devices made of MAPbI$_3$ single crystal with different architectures (see Supplementary Fig. 1 and Supplementary Fig. 2 for material characterizations). The #1 device has an Au/MAPbI$_3$/Au structure (shortened as Au/Au), whereas the #2, #3, and #4 device have Pb/MAPbI$_3$/Au structure (shortened as Pb/Au). The dimensions of the devices are listed in Supplementary Table 1. The surface of MAPbI$_3$ single crystal is typically p-type[22], which results in Ohmic and Schottky junction when forming contact with Au and Pb, respectively[10]. The Pb/Au devices show a clear current rectifying behavior with a small reverse saturation current (electric field direction under reverse bias is from Pb to Au) (Fig. 3a, b). In comparison, the Au/Au device is a double Ohmic structure, showing a negligible current rectifying behavior and hence a large dark current under both biasing directions (Fig. 3b). According to the *I–V* behavior, the

Pb/MAPbI$_3$/Au architecture can be considered as photodiode structure while the Au/MAPbI$_3$/Au architecture is considered as photoconductor structure.

The energy band diagram of a reversely biased Pb/Au device is shown in Supplementary Fig. 3, showing that the charge injection from metal electrode into perovskite is negligible due to the energy barrier at the metal-semiconductor interface, which results in the known fact that the photoconductive gain for a reversely biased photodiode is at most 1 because no charge carrier re-circulating could happen due to the negligible charge injection into the semiconductor[37,38]. For a forward biased Pb/Au device, charge carrier can flow freely from metal electrode into perovskite due to the negligible energy barrier at metal-semiconductor interface (see Supplementary Fig. 3 for energy band diagram), which results in the photoconductive gain >1 as charge carrier re-circulating is possible[37,38]. The photoconductive gain for the Au/Au structure could also be larger than 1 because of the negligible energy barrier at metal-semiconductor interface (Supplementary Fig. 3).

A larger photoconductive gain due to charge carrier re-circulating means more induced charge on electrode for the same amount of generated charge carriers. As expected, the sensitivity of the forward biased Pb/Au device and the Au/Au device is ~1–2 orders of magnitude larger than the sensitivity of the reversely biased Pb/Au device (Fig. 3c–e) (see Supplementary Fig. 4 and Supplementary Fig. 5 for sensitivity calculation). Our experiments show that the device architecture and the operational mode play a critical role in controlling the sensitivity, which, however, is often overlooked when sensitivity values are being reported for perovskite X-ray detectors. Some of the reported sensitivity value summarized in Fig. 1 reach as high as $1.22 \times 10^5$ μC Gy$_{air}^{-1}$ cm$^{-2}$, but a giant sensitivity does not necessarily indicate a superior material or device quality without an equitable ground of detector architecture and operation mode.

**Calculation of MAPbI$_3$ X-ray dose rate detection limit**. We demonstrate mathematically and compare detection limit $\dot{D}_{limit}$ values obtained by both the dark current ($I_{dark}$ & $s$) method and the X-ray photocurrent ($I_{dark}$ & $I_{X-ray}$) method. The lowest X-ray dose rate that can be generated by our experimental setup is 5 nGy$_{air}$ s$^{-1}$ and increases to 12 nGy$_{air}$ s$^{-1}$ with an Aluminum attenuator removed. The photocurrent under X-ray $I_{X-ray}$ is more eminent at dose rate of 12 nGy$_{air}$ s$^{-1}$ than that at 5 nGy$_{air}$ s$^{-1}$ for #2 Pb/Au device under reverse 2 V (Fig. 4a), and it can be expected that $I_{X-ray}$ will be indiscernible from dark current by visual inspection if the dose rate is further reduced. Following the dark current ($I_{dark}$ & $s$) method, we calculate a conservative detection limit of net current $I_{limit} = \Delta = 3.29\sigma_{I_{dark}}$ with preset value of $n_{I_{X-ray}} = 1$ and $n_{I_{dark}} \gg n_{I_{X-ray}}$. We collected a large number of digitized current data points (typically ~ 500) for $I_{dark}$ to satisfy that $n_{I_{dark}}$ is very large. The dark current standard deviation, i.e., $\sigma_{I_{dark}}$, is ~0.376 pA calculated using the dark current that has a minor drifting after a long time of biasing, which yields $I_{limit} = 1.24$ pA. After conversion, $\dot{D}_{limit}$ is calculated to be $\dot{D}_{limit} = I_{limit}/(A * s) = 2.4$ nGy$_{air}$ s$^{-1}$, where the electrode area is $A = 0.0468$ cm$^2$ and the sensitivity is $s = 11{,}180$ μC Gy$_{air}^{-1}$ cm$^{-2}$ under reverse 2 V of the #2 Pb/Au device.

To obtain $\dot{D}_{limit}$ by the X-ray photocurrent ($I_{dark}$ & $I_{X-ray}$) method, we lowered the X-ray dose rate successively and check if each dose rate is detected. We show below that the lowest achievable 5 nGy$_{air}$ s$^{-1}$ is detected under reverse 2 V of the #2 Pb/Au device. We collected a large number of digitized current data points (typically ~ 500) for the dark current to make $n_{I_{dark}}$ very large so that $\sigma_{I_{dark}}^2/n_{I_{dark}}$ is negligible. We have $n_{I_{X-ray}} = 17$ data points for a specific $I_{X-ray}$ measurement under 5 nGy$_{air}$ s$^{-1}$. The

**Table 2 # 2 Pb/Au device detection limit of X-ray dose rate $\dot{D}_{limit}$ determined by dark current ($I_{dark}$ & $s$) method vs by X-ray photocurrent ($I_{dark}$ & $I_{X-ray}$) method.**

|  | $\dot{D}_{limit}$ (nGy$_{air}$ s$^{-1}$) by $I_{dark}$ & $s$ method | $\dot{D}_{limit}$ (nGy$_{air}$ s$^{-1}$) by $I_{dark}$ & $I_{X-ray}$ method |
|---|---|---|
| 0 V | 4.9 | 12 |
| Reverse 2 V | 2.4 | 5 |
| Reverse 5 V | 2.7 | 5 |
| Reverse 10 V | 6.9 | 24 |
| Reverse 20 V | 14.6 | 61 |
| Forward 2 V | 77.1 | 150 |
| Forward 10 V | 45.8 | 150 |

standard deviation of $I_{X-ray}$, i.e., $\sigma_{I_{X-ray}}$, is calculated to be ~ 0.468 pA. Then we can calculate a detection limit for this specific measurement to be $\Delta = (z_{1-\alpha} + z_{1-\beta})\sqrt{\sigma_{I_{X-ray}}^2/n_{I_{X-ray}}} = 0.37$ pA with $z_{1-\alpha} = z_{1-\beta} = 1.645$. Meanwhile, the measured difference of the mean dark current and the mean photocurrent under X-ray, i.e., $|\mu_{I_{X-ray}} - \mu_{I_{dark}}|$, is 5.9 pA. As $|\mu_{I_{X-ray}} - \mu_{I_{dark}}| > \Delta$ for this specific measurement, the dose rate of 5 nGy$_{air}$ s$^{-1}$ is considered as detected, satisfying $z_{1-\alpha} = z_{1-\beta} = 1.645$.

As expected, the X-ray photocurrent ($I_{dark}$ & $I_{X-ray}$) method yields a larger $\dot{D}_{limit}$ than that from the dark current ($I_{dark}$ & $s$) method, which is listed in Table 2 for a quantitative comparison (see Supplementary Fig. 6 for detector current response to the lowest detectable dose rate).

Following the $I_{dark}$ & $s$ method, the $I_{limit}$ and $\dot{D}_{limit}$ of the #2 Pb/Au device at different reverse bias voltage were obtained (Fig. 4b). The $I_{limit}$ increases as reverse bias voltage increases because the increased reverse bias leads to the increased mean value and the standard deviation of the dark current. Although $I_{limit}$ monotonically increases as function of reverse bias, $\dot{D}_{limit} = I_{limit}/(A * s)$ dose not necessarily increases monotonically since the sensitivity $s$ also increases as reverse bias increases (Fig. 3c). Compared to the reversely biased #2 Pb/Au devices, the forward biased #2 Pb/Au device has a significantly higher $I_{limit}$ (Fig. 4c) because of the large dark current under forward bias. The #1 Au/Au device has $I_{limit}$ at the same level as that of the forward biased #2 Pb/Au device. Despite that $I_{limit}$ increased dramatically by ~1000 times for the forward biased #2 Pb/Au device, the $\dot{D}_{limit}$ only increased by ~10 folds according to $\dot{D}_{limit} = I_{limit}/(A * s)$, because the sensitivity of the forward biased #2 Pb/Au device are ~2 orders of magnitude higher than the reversely biased #2 Pb/Au devices. A quantitative comparison of the sensitivity and $\dot{D}_{limit}$ of #2 Pb/Au device under reverse vs forward bias mode is presented in Table 3.

To present a complete picture of device architecture and operation mode's influence on sensitivity and detection limit $\dot{D}_{limit}$, we qualitatively illustrate the current value as a function of applied bias voltage for a photodiode working under forward vs reverse bias mode in Fig. 4d. Although the photodiode under forward bias mode has a larger sensitivity (slope of the linear fitting) than that under reverse bias mode, the forward bias mode also has a larger dark current. The X-ray dose rate cannot be detected infinitesimally. Instead, the X-ray dose rate detection limit $\dot{D}_{limit}$ is depending on the detection limit of net current $I_{limit}$ and the device sensitivity $s$, and $I_{limit}$ is further determined by the dark current. The specific quantitative relationship between $I_{limit}$ and the dark current is established by the statistical model in this paper.

In summary, based on a well-established statistical model and equations, we propose processes for the detection limit

determination of X-ray detectors, including the dark current ($I_{dark}$ & $s$) method by the measurement of dark current $I_{dark}$ and sensitivity $s$, and the X-ray photocurrent ($I_{dark}$ & $I_{X-ray}$) method through repetitive measurements of $I_{dark}$ and photocurrent under X-ray irradiation $I_{X-ray}$. Mathematically, our method considers the different sample size (i.e., number of sampled data points) and possibly different standard deviation of both blank and gross signal, which reduces to the subset of Currie's formulars with certain assumptions on the sample sizes and standard deviations are met (see Table 4).

The X-ray photocurrent ($I_{dark}$ & $I_{X-ray}$) method requires successively lowering the X-ray dose rate. In comparison, the dark current ($I_{dark}$ & $s$) method only needs measurement of dark current and sensitivity, which is easy to perform experimentally. A practical procedure of both methods is provided below in Fig. 5 for an easy adoption.

We also demonstrate the critical role of device architecture and operation mode in controlling the perovskite X-ray detector sensitivity and detection limit. When comparing sensitivity of different perovskite X-ray detectors for material quality and device performance comparison, it is important to highlight the device architecture and working mode for a fair competition. Our work could facilitate effective design and characterization of perovskite X-ray detectors for medical imaging or other non-

perovskite photoelectric sensor systems. When the development of perovskite X-ray detectors is advanced to the pixelated imager with a single readout of X-ray scan, the system level characterization metrics for spatial resolution and noise performance such as modulation transfer function, noise power spectrum, and detective quantum efficiency may be better suited for their characterizations at that stage.

## Methods

**Crystal growth.** MAPbI$_3$ thin single crystals were grown using the space confined inverse temperature crystal growth method as reported in literatures[39,40]. Briefly, an equimolar ratio of methylammonium iodide (MAI) and lead iodide (PbI) are dissolved in γ butyrolactone (GBL) to obtain a 1.2 M solution. By heating 2 mL of this solution at 140 °C for a few hours, a small MAPbI$_3$ crystal (<1 mm) is obtained. One small crystal is used as a seed and placed inside a fresh solution and the crystal continues to grow into larger size. The seed crystal in the cavity is allowed to grow for 3 days in a fresh precursor solution kept at 90 °C to obtain large (~1 cm) thin MAPbI$_3$ single crystals. The crystals are then taken out and washed in Toluene. Finally, we polish the surface of the crystal with SiC sandpaper followed by diamond paste polishing with grit size 3 μm, 1 μm, and 0.25 μm, sequentially.

**Detector fabrication.** The polished thin MAPbI$_3$ crystals are transferred in an Argon filled glovebox for device fabrication. For the hole selective contact, 50 μl of PTAA (10 mg ml$^{-1}$ in CB) is spin-coated at 3000 rpm for 40 s on one side of the flat crystal followed by 100 nm of Au deposition using e-beam. For the electron selective contact, 50 μl of PCBM (20 mg ml$^{-1}$ in CB) is spin-coated at 3000 rpm for

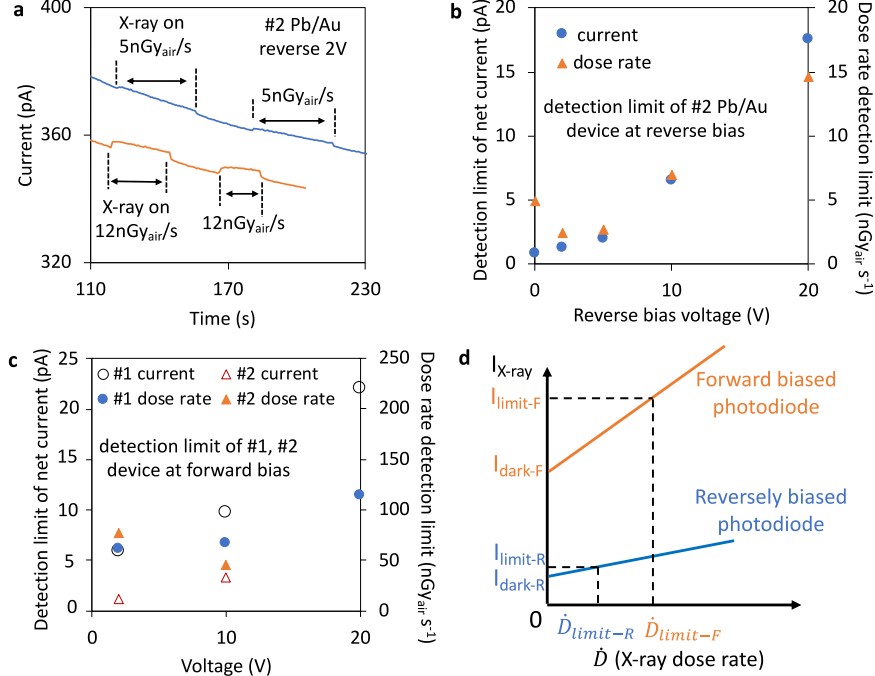

**Fig. 4 Detection limit obtained by the dark current method for MAPbI$_3$ detectors working in different mode. a** Experimentally measured current response of reversely biased #2 Pb/Au photodiode to X-ray dose rates approaching the detection limit. Calculated detection limit of net current $I_{limit}$ and X-ray dose rate detection limit $\dot{D}_{limit}$ for the **b** reversely biased #2 Pb/Au photodiode, and for the **c** #1 Au/Au photoconductor and forward biased #2 Pb/Au photodiode. **d** qualitative comparison of sensitivity (slope of the linear fitting) and dark current for photodiode under forward vs reverse bias mode. The subscripted F and R stands for forward and reverse bias mode, respectively.

**Table 3 # 2 Pb/Au device detection limit $\dot{D}_{limit}$ and sensitivity under reverse vs forward bias mode.**

|  | Reverse 2 V | Forward 2 V | Reverse 10 V | Forward 10 V |
|---|---|---|---|---|
| Detection limit $\dot{D}_{limit}$ (nGy$_{air}$ s$^{-1}$) | 2.4 | 77.1 | 6.9 | 45.8 |
| Sensitivity (μC Gy$_{air}^{-1}$ cm$^{-2}$) | $1.1 \times 10^4$ | $3.5 \times 10^5$ | $2.0 \times 10^4$ | $1.5 \times 10^6$ |

**Table 4 The mathematical reduction of Detection Limit equations to Currie formulars[31] under certain assumptions.**

| Detection Limit equations | $n_2 = n_1$ | $n_2 \gg n_1$ |
|---|---|---|
| $\Delta = \lvert \mu_1 - \mu_2 \rvert = (z_{1-\alpha} + z_{1-\beta})\sqrt{\frac{\sigma_1^2}{n_1} + \frac{\sigma_2^2}{n_2}}$ | $\Delta = 4.65 \frac{\sigma_2}{\sqrt{n_1}}$ | $\Delta = 3.29 \frac{\sigma_2}{\sqrt{n_1}}$ |
| | $\Delta = 4.65\sigma_2$ (reduce to Paired observations[31] when $n_1 = 1$) | $\Delta = 3.29\sigma_2$ (reduce to "Well-known" blank[31] when $n_1 = 1$) |

Assumptions and definitions: (1) gross signal standard deviation $\sigma_1$ = blank signal standard deviation $\sigma_2$. (2) $z_{1-\alpha}$ = 1.645 and $z_{1-\beta}$ = 1.645 are the Z-score corresponding to type I error $\alpha$ = 5% and type II error $\beta$ = 5%, respectively, for one-sided test. (3) $n_1$, $\mu_1$ and $n_2$, $\mu_2$ are the sample size and mean value of gross signal and blank signal, respectively. $\Delta$ is the detection limit.

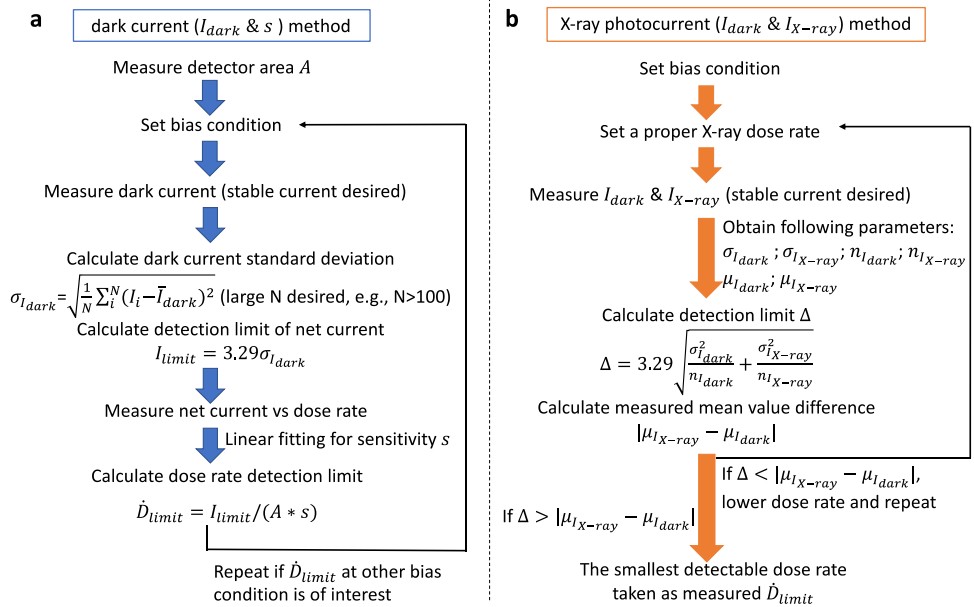

**Fig. 5 A practical procedure for detection limit determination. a** The dark current $I_{dark}$ & s method. N is the number of digitized current points of dark current and $I_i$ is the i[th] point. **b** The X-ray photocurrent $I_{dark}$ & $I_{X-ray}$ method. $\sigma_{I_{X-ray}}$ is the standard deviation of the photocurrent under X-ray irradiation $I_{X-ray}$. $n_{I_{X-ray}}$ and $n_{I_{dark}}$ are the number of digitized current points of $I_{X-ray}$ and $I_{dark}$, respectively. $\mu_{I_{X-ray}}$ and $\mu_{I_{dark}}$ are the measured mean value of $I_{X-ray}$ and $I_{dark}$, respectively.

40 s on the opposite side followed by thermal deposition of 5 nm BCP and 100 nm of Pb. The hole/electron selective contacts do not affect the overall Schottky or Ohmic contact behavior of the devices and may help improve the barrier height to suppress charge injection from metal electrode into MAPbI₃.

**Detector measurement setup.** The X-ray beam is generated by an X-ray tube with Ag target (Amptek Mini-X X-ray tube) (see Supplementary Fig. 7a for X-ray energy spectrum). The device was located at the axis of the X-ray beam with a distance ~20 cm where the X-ray dose rate in air was carefully calibrated by a dosimeter (Fluke Biomedical RaySafe 452). For all measurements, the X-ray tube is operated at a constant voltage of 30 kV. The X-ray dose rates in air at X-ray tube voltage of 30 kV and varying X-ray tube current are shown in Supplementary Fig. 7b. For detection limit measurements, thin Aluminum sheets were added as attenuators between the device and the X-ray tube to reduce the X-ray dose rate in air. The device current and voltage were measured by Keithley 4200A-SCS parameter analyzer. The sensitivity of all MAPbI₃ detectors were measured by collecting hole dominantly induced signal, e.g., X-rays irradiating the Pb electrode when the Pb/Au device is reversely biased, as the X-ray photons have limited penetration depth in MAPbI₃ (see Supplementary Fig. 8).

**Normal distribution of the X-ray detector current.** The histogram of current counts vs current is shown in Supplementary Fig. 9. The dark current of a Pb/Au device at 0 V was measured to eliminate the interference of dark current drift at non-zero bias voltage. Experimentally measured current distribution supports the fundamental mathematical assumption that X-ray detector current follows Normal distribution.

## Data availability
The data that support the plots within this paper and other findings of this study are available from the corresponding author upon reasonable request.

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

## Acknowledgements
This work was supported by the U.S. Department of the Defense, Defense Threat Reduction Agency under Grant HDTRA1-19-1-0024 and partially supported by the U. S. Department of Energy/National Nuclear Security Administration under Award Number DE-NA0003921. W.N. and S.S. acknowledges the Los Alamos National Laboratory's LDRD Mission Foundation Project. Their work was performed, in part, at the Center for Integrated Nanotechnologies, an Office of Science User Facility operated for the U.S. Department of Energy, Office of Science by Los Alamos National Laboratory (Contract 89233218CNA000001) and Sandia National Laboratory (Contract DE-NA-0003525). We thank Miss. Qingyu Chen, Ph.D. student at The Ohio State University, for the helpful discussion of the relevant statistical theory.

## Author contributions
L.R.C. and L.P. conceived the idea and proposed the statistical method. L.P. designed experiments and characterized detector performance. W.N. and S.S. synthesized, characterized the single crystals and fabricated the devices. N.T. contributed to the discussion of detection limit. L.P., L.R.C., and W.N. wrote the manuscript. All authors discussed the results and commented on the manuscript.

## Competing interests
The authors declare no competing interests.
