## [Peer Review File · Nature Communications]

Determination of X-ray detection limit and applications in perovskite X-ray detectorsEditorial Note: Parts of this Peer Review File have been redacted as indicated to remove third-party material where no permission to publish could be obtained.

Reviewers' comments:

Reviewer #1 (Remarks to the Author):

The paper addresses a question which I find important and highly relevant in spite of the number of papers which currently appear about perovskite X-ray detectors. The question is, how to properly determine the detection limit, which is one of the most important characteristic parameters of an X-ray detector. On the other hand, the paper presents rather a technical detail, than an important break-through, makes several statements which are possibly not completely correct, is rather difficult to read due to numerous new vocabularies which are used or misused, and finally fails to convincingly show that the proposed method is accurate or of any practical use. Thus, I rather doubt that this paper will be useful for a broad readership. The given procedure for detection limit determination is rather useful as a guideline for a certification laboratory, which will be hopefully introduced as a rule for all further X-ray detection papers claiming record detection limits.

Good points in this paper are that it is shown that the detection limit is not simply a given number for a detector, but it depends also on the bias condition. In the previous papers so far only the record value for the detection limit was provided, which is most probably also the most important number, because the detectors will be used under optimized conditions.

What I find misleading:

In the abstract there is a statement saying that a prior detection limit can be calculated through only dark current measurement. In line 67, in contrast it says that a calibration factor is needed in order to calculate the prior detection limit. And this calibration factor is the sensitivity of the perovskite X-ray detector. The sensitivity, however, is not measured from the dark current but under X-ray illumination.

In line 70 it is stated that the prior detection limit is consistent with the experimentally determined detection limit by posterior check of the detectability. In line 294 the calculated detection limit is given as 2.4 and the "measured" one is however 5 nGyair/s which represents a 100% error.

In 136 it is stated that there is a discrepancy between IUPAC method and practiced method in that the noise current is taken from the standard deviation of the "cross signal". Honestly, I think that no one is doing that, but the noise is measured as the standard deviation of the dark current in disagreement with the formula after line 130.

In 177 it is stated that the sample size of the cross signal, i.e., cross current with X-ray on, is better to be kept as small as needed to reduce the cumulative X-ray dose to patients. This statement is really misleading because no one in the world is measuring the detection limit of his detector in the presence of any patients. There are no increasing costs related to do sufficiently long measurements for characterizing a detector under X-ray illumination properly, as is stated in line 175.

Honestly, I wonder, what would be the detection limit for the presented detector, when the usual practical procedure would be used, the signal to noise ratio should be higher than 3. Is that totally different than the provided values from the new methods?

Personally, I dislike the some of the vocabularies used to describe the detector operation in the experimental part. The operation of the photoconductor is called "charge injection" mode. Why is that, are the gold contacts not forming and ohmic contact? In reverse direction the description uses a "suppression" of dark current. However, in the ideal diode equation the current in reverse direction is not called "suppressed" current but rather reverse saturation current. What I find also an unlucky naming is what is described here as "sample size effect". The sample size could also mean in this manuscript the dimensions of the perovskite specimens.

Thus, in total I find this manuscript in the present form rather not suitable for publication in Nature Communications.

Reviewer #2 (Remarks to the Author):

In this manuscript, the authors reported a different method to determine the detection limit of X-ray detectors. This is a publishable result, but not for Nature Communications, since it fits better in a specialized journal for the reasons listed below.

The difference of the detection limit is most likely caused by the material quality of contact, rather than the method of characterization.

The method shown here does not provide direct guidance in X-ray imaging application where the algorithm of imaging collection and process can be very different. What is shown in literature so far does not represent the real imaging process either, but the method represented here did not improve much.

The whole content does not have to be related to perovskites.

The time scale in Figure 4a is not comparable to real x-ray imaging.

Reviewer #3 (Remarks to the Author):

The authors well studied determination of x-ray detection limit for various perovskite X-ray detectors. The whole study is highly important for this field and the result could help build discipline to quantitatively judge the performance. There are still several issues to be addressed.

1. The charge collection and injection mode could result in totally different dark current and sensitivity. The prior detection limit seems suitable for charge collection mode. However, for charge injection mode, the photoconductive gain effect will increase the noise value (also known as generation-recombination noise) for signal current. Thereby, it will not correct to use the dark current to calculate the detection limit for collection mode. The detection limit for #2 at reverse/forward bias should be quantitatively compared, not in the level of orders.
2. Moreover, according to Figure 4a, the dark current is drifting in the measurement period. Then how to determine the exact dark current value. Also, will the ionic drifting in perovskite influence the detection limit. For perovskite detectors, the ionic drifting current is sometimes increasing and sometimes decreasing, which will have totally different impact on the dark current value. But the ionic drifting is not involved in their equation.
3. The authors should give some comments on what parameters could best express the performance of perovskite detectors. High sensitivity or low detection limit?
4. Figure 1a summarizes the sensitivity and detection limit for reported results. Are these results following the prior detection limit value? The author should give a check, which is important for the general applicability of this paper.

Response to Reviewer Comments:

Dear Editor and Reviewers,

We sincerely thank you all for carefully reviewing our manuscript and providing your valuable suggestions. We believe that we have made every effort to address all your comments. In the following, please find our individual responses to each comment including the changes we have made in the original manuscript.

The reviewers' comments are in 'bold' font, while our responses are in regular font. Our revisions in the manuscript are highlighted in yellow.

Reviewer #1 (Remarks to the Author):

The paper addresses a question which I find important and highly relevant in spite of the number of papers which currently appear about perovskite X-ray detectors. The question is, how to properly determine the detection limit, which is one of the most important characteristic parameters of an X-ray detector. On the other hand, the paper presents rather a technical detail, than an important break-through, makes several statements which are possibly not completely correct, is rather difficult to read due to numerous new vocabularies which are used or misused, and finally fails to convincingly show that the proposed method is accurate or of any practical use. Thus, I rather doubt that this paper will be useful for a broad readership. The given procedure for detection limit determination is rather useful as a guideline for a certification laboratory, which will be hopefully introduced as a rule for all further X-ray detection papers claiming record detection limits.

We sincerely appreciate the reviewer's comments.

Research efforts on perovskite X-ray detectors have seen surged publications in recent years. Although, as the reviewer said, the detection limit is one of the most important characteristic parameters of an X-ray detector, the methodology on how to properly determine the X-ray detection limit has not been able to catch up with the rapid development of perovskite material synthesizing methods.

We believe the ambiguity and the misleading concepts on the way how people determine X-ray detection limits, including those published in high impact journals, call for the urgent needs to shed clarifications on this topic to the community. As reviewer #3 suggested, to improve "the general applicability of this paper", we made revision to the manuscript to list the method of detection limit determination for each individual paper summarized in **Figure 1**. The explicit listing of how numerous papers dealing with detection limit clearly shows the current ambiguity of methodology for X-ray detection limit determination.

Please see changes in Page 2 Line 36

"Many publications^{12-14,16-19} have only reported the sensitivity of their perovskite X-ray detectors, while lacking the measure of the detection limits (Figure 1). Among those that have reported the detection limit (Figure 1), some papers^{7,9,20} did not present clearly the method for their detection limit determination, and some²¹⁻²⁵ claimed the use of a method based on the 1975 International Union of Pure and Applied Chemistry (IUPAC) detection limit definition²⁶ without an explicit

equation provided. Although some of the work²⁷⁻²⁹ have also adopted a method referenced as derivatives of the IUPAC definitions, there has been no discussions on its statistical validity.”

The practiced method with explicit equations that can be found in papers [27–29] is claimed as based on the 1975 IUPAC definition of the detection limit. The IUPAC definition is further descended from Currie method published 1968. However, the Currie method, hence the following IUPAC definition deals with different physics compared to the X-ray current detection. Please see our changes in Page 7 Lines 140-149 for a detailed analysis.

“A simple modification to the IUPAC definition would not result in a correct equation to use for detection limit determination because of the different physics involved. IUPAC definition follows the Currie method that is applicable for γ -ray photon counting where the physics behind is radioactive decay. However, in X-ray detection, the large quantity of X-ray photons emitted from X-ray machine is not a decay phenomenon in nature, nor does the dark current and the gross current of an X-ray detector. The statistical distribution applicable to photon counting is Poisson-Normal distribution where the sample size n that refers to number of counting performed, is typically 1 or not far larger than 1. Contrarily, the dark current and gross current for an X-ray detector follows Normal distribution, where the sample size n in the current measurement referred as the number of digitized current points is typically very large.”

In this paper, we introduced a statistical model that fits into the physics of the X-ray induced photocurrent measurement. Based on such model, we proposed a method that is fundamentally new and applicable to X-ray detector working in a current mode as opposed to γ -ray photon counting problems. We believe our model and method help to shed lights on the current ambiguity and maybe adopted to form a new standard for X-ray detector detection limit determination.

As reviewer #2 commented “**The whole content does not have to be related to perovskites**”. We agree, the methodology is also applicable to other X-ray detectors, or even photodetectors. The citations to the publications (two examples given at below) on demonstrating perovskite’s X-ray detection capability has been high and growing fast, from which we do believe our publication would have a broad and diverse readership.

Wei, Haotong, et al. "Sensitive X-ray detectors made of methylammonium lead tribromide perovskite single crystals." *Nature Photonics* 10.5 (2016): 333-339.

- **8930** Accesses
- **632** Citations

Kim, Yong Churl, et al. "Printable organometallic perovskite enables large-area, low-dose X-ray imaging." *Nature* 550.7674 (2017): 87-91.

- **12k** Accesses
- **324** Citations

Our new method is useful as a guideline for a certification laboratory, and we also believe that our method might be broadly adopted and cited as X-ray detection limit characterization method by incoming papers. For easy adoption, we provide a practical procedure to characterize and compare X-ray detection limits at the end of the paper. Please see changes in Page 19 Line 361.

“A practical procedure of both *prior* calculation and *posterior* check is provided below in Figure 5 for an easy adoption.

Figure 5. a. practical procedures of theoretical detection limit determination by *prior* calculation **b.** practical procedures of detection limit experimental measurement by *posterior* check. $\sigma_{I_{gross}}$ is the standard deviation of the gross current measured with X-ray. $n_{I_{gross}}$ and $n_{I_{dark}}$ are the number of digitized current points of gross current and dark current, respectively. $\mu_{I_{gross}}$ and $\mu_{I_{dark}}$ are the measured mean value of gross current and dark current, respectively.

We appreciate that the reviewer brought up the point that “our paper is rather difficult to read due to numerous new vocabularies”. We believe this is not a surprise for an interdisciplinary research since this paper involves photonics, X-ray detection, and statistical terminologies. We have made every effort to give context where terminologies are being introduced. For example, we have clarified sample points at where it means “number of data points” or “digitally sampled current data points”.

Good points in this paper are that it is shown that the detection limit is not simply a given number for a detector, but it depends also on the bias condition. In the previous papers so far only the record value for the detection limit was provided, which is most probably also the most important number, because the detectors will be used under optimized conditions.

Thanks for the comment. As reviewer #3 commented, to “best express the performance of perovskite detectors”, we made revision by adding a figure (**Figure 4d**) and a table (**Table 3**) to show qualitatively/quantitatively the device operation condition’s influence, e.g., reversely vs forward biased photodiode, on detection limit and sensitivity.

Please see Page 18 Lines 332-340.

“To present a comprehensive picture of device architecture and operation mode’s influence on the sensitivity and the detection limit, we qualitatively illustrate the current value as a function of applied bias voltage for a photodiode working under forward vs reverse bias mode in **Figure 4d**. Although the photodiode at forward bias mode has a larger sensitivity (slope of the linear fitting) than that under reverse bias mode, the forward bias mode also has a larger dark current. The X-ray dose rate cannot be detected infinitesimally. Instead, the X-ray dose rate detection limit \dot{D}_{limit} is depending on the photocurrent detection limit I_{limit} and the device sensitivity s , and I_{limit} is further determined by the dark current. The specific quantitative relationship between I_{limit} and the dark current is established by the statistical model in this paper.”

Figure 4d. qualitative comparison of sensitivity (slope of the linear fitting) and dark current for photodiode working under forward vs reverse bias mode.

Please see Page 17 Line 327.

“A quantitative comparison of the sensitivity and detection limit of #2 Pb/Au device under reverse vs forward bias mode was presented in **Table 3**.”

Table 3. #2 Pb/Au device detection limit and sensitivity under reverse vs forward bias mode

	Reverse 2V	Forward 2V	Reverse 10V	Forward 10V
Detection limit (nGy _{air} /s)	2.4	77.1	6.9	45.8
Sensitivity (μC/Gy _{air} /cm ²)	1.1×10 ⁴	3.5×10 ⁵	2.0×10 ⁴	1.5×10 ⁶

What I find misleading:

In the abstract there is a statement saying that a prior detection limit can be calculated through only dark current measurement. In line 67, in contrast it says that a calibration factor is needed in order to calculate the prior detection limit. And this calibration factor is the sensitivity of the perovskite X-ray detector. The sensitivity, however, is not measured from the dark current but under X-ray illumination.

We admit that we could have made our case better and clear. Yes, we need to use sensitivity as a calibration factor to derive the detection limit of X-ray dose rate \dot{D}_{limit} . It is the detection limit of photocurrent I_{limit} that can be calculated through only dark current measurement,

To clarify, we have made a revision to associate the term “detection limit” with either “photocurrent” or “X-ray dose rate” at where it appears, so they are well defined within the context in reminding readers of their specific meaning.

Please see changes in Page 1 Lines 17-19.

“A *prior* detection limit can be calculated through the measurement of dark current and sensitivity as a calibration factor, which yields the theoretical detection limit with an easy-to-follow practice.”

Please see changes in Page 12 Lines 229-231.

“The measurement of the dark current can produce *prior* calculated detection limit of photocurrent I_{limit} . To convert I_{limit} to the detection limit of X-ray dose rate \dot{D}_{limit} (interchangeably, the lowest detectable dose rate), we need a calibration factor^{31,35}, ...”

In line 70 it is stated that the prior detection limit is consistent with the experimentally determined detection limit by posterior check of the detectability. In line 294 the calculated detection limit is given as 2.4 and the “measured” one is however 5 nGy_{air}/s which represents a 100% error.

Thanks for the comment. It’s true that the “measured” value, 5 nGy_{air}/s, is 2 times higher than the “calculated” detection limits at 2.4 nGy_{air}/s.

One intuitive but improper way of measuring X-ray detection limit is to lower the X-ray dose rate as much as possible to determine the photocurrent response of the perovskite detector. While being a tedious process to repeatedly measure the current signal by reducing X-ray dose rate approaching to the background, the experimentally determined detection limit is asymptotic that can never equal to the true value of the detection limits. The lower it goes, the higher the uncertainty of the dose rate. Essentially one is measuring the dark current of the perovskite device under natural radiation background. Then the question arises: how to determine the detection limit by only measuring the dark current? How to validate it with a measured value when X-ray machine is powered on? How to evaluate those reported lowest X-ray dose rate when different groups are using different models of X-ray machine? Is there a consistent statistical model to consolidate the predetermined X-ray detection limit (*prior* calculation) with the measured value (*posterior* check)?

Our paper is exactly addressing this unsolved ambiguity by proposing a statistical model for determining the *prior* calculated detection limits and then validate it with the *posterior* check.

The *posterior* check requires one to decrease the X-ray dose rate successively for each measurement. The *prior* calculation can yield the true theoretical statistical limit that can never be

equal by the *posterior* check method. Hence, the theoretical detection limit of X-ray dose rate determined by *prior calculation* method is 2.4 nGy_{air}/s, which is lower than the 5 nGy_{air}/s experimentally determined value. The difference is well expected.

Please see our changes in Page 11 Lines 211-214 for details.

“Following the principles above, the *posterior* check method can be performed by successively lowering the X-ray dose rate used to generate the gross current until the gross current cannot be detected against the dark current. The lowest X-ray dose rate that generates the smallest detectable gross current is taken as the X-ray dose rate detection limit.”

Please see our changes in Page 11 Line 216 for details.

“Apparently, the experimental procedures of the *posterior* check method require a repetitive and tedious work, and it can only produce an asymptotic value that will always be larger than the true theoretical detection limit obtained from the *prior* calculation method. Furthermore, the *posterior* check approach could yield a large uncertainty of the measured detection limit value due to the limitation of the experiment instruments, *e.g.*, X-ray tube, number of attenuators used. Contrarily, the *prior* calculation of the theoretical detection limits only needs a measurement of the dark current and device X-ray sensitivity as a calibration factor.”

Simply put, the 2.4 nGy_{air}/s only depends on perovskite device, while 5 nGy_{air}/s depends on models of the X-ray machine and experimental and other human factors.

In 136 it is stated that there is a discrepancy between IUPAC method and practiced method in that the noise current is taken from the standard deviation of the “cross signal”. Honestly, I think that no one is doing that, but the noise is measured as the standard deviation of the dark current in disagreement with the formular after line 130.

Thanks for the comment. We have double checked the equation provided in the practiced method used in ref [27–29]. We believe, to the best of our understanding, they are taking noise current from the standard deviation of “photocurrent” that is named as “gross signal” or “gross current” in our paper. Please see below for what we found.

We quoted the original text and equations from one of the papers using the practiced method, which is shown below. The term “photocurrent” in their paper is equivalent to the term “gross current” in our paper. We use different terms in our paper because we think “gross current” represents the current measured with X-ray turned on, while net photocurrent is the difference between “gross current” and “dark current”. Their equation indicates that they take noise current as the standard deviation of the “photocurrent” named in their paper (or “gross current” named in our paper).

Please see quoted original text from “*Nature photonics* 11.11 (2017): 726-732.”,

[REDACTED]

From their explanation, we further believe the reason why the practiced method in ref [27-29] did not take noise current as the standard deviation of dark current is that they found the standard deviation of the dark current is different from that of “gross current”. The initiative to consider such difference is applaudable, but simply replacing the standard deviation of the dark current by the standard deviation of the “gross current” lacks a statistical validity.

The correct statistical model to consider these factors are the Detection Limit equations we presented. Please see our revision Page 7 Lines 129-137 for a detailed discussion.

“ In the IUPAC definition, the noise value is taken as the standard deviation of the blank signal, σ_B , as it follows the Currie method assuming the standard deviation is the same for blank signal and for gross signal, that is, $\sigma_B = \sigma_Q$. However, practically, the noise current I_{noise} taken as the standard deviation of gross current $\sigma_{I_{gross}}$ will typically be larger than the noise current taken as the standard deviation of dark current $\sigma_{I_{dark}}$ due to the fluctuation of X-ray photon flux, X-ray generated charge carrier generation and recombination. The practiced method²⁷⁻²⁹, shown in **Figure 2b**, considered such difference by simply replacing $\sigma_{I_{dark}}$ in the IUPAC definition with $\sigma_{I_{gross}}$, which lacks statistical validity and results in an improper equation to use.”

In 177 it is sated that the sample size of the cross signal, i.e., cross current with X-ray on, is better to be kept as small as needed to reduce the cumulative X-ray dose to patients. This statement is really misleading because no one in the world is measuring the detection limit of his detector in the presence of any patients. There are no increasing costs related to do sufficiently long measurements for characterizing a detector under X-ray illumination properly, as is stated in line 175.

We agree. We have removed the statement about the cost of instrument calibration from the manuscript.

Honestly, I wonder, what would be the detection limit for the presented detector, when the usual practical procedure would be used, the signal to noise ratio should be higher than 3. Is that totally different than the provided values from the new methods?

We appreciate the constructive comment. A comparison of the detection limit values from different methods is helpful to show their difference quantitatively.

We compared the measured detection limit values by our *posterior* check method and the theoretical detection limit values by our *prior* calculation, which are listed as a revision in **Table 2**.

The measured detection limit values by *posterior* check that has the same procedures as that in the practiced method [27-29] are all higher than the theoretical detection limit values by *prior* calculation, which reflects the large uncertainty of the usual practical procedures due to limitation of experiment instrument and other human factors.

Please see below the revised text at Page 16 Line 309.

“As expected, the *posterior* check yields a larger detection limit value than that from the *prior* calculation, which is listed in **Table 2** for a quantitative comparison (see **Figure S6** for detector current response to the lowest detectable dose rate).”

Table 2. #2 Pb/Au device theoretical detection limit by *prior* calculation vs measured detection limit by *posterior* check

	Theoretical detection limit by prior calculation (nGy _{air} /s)	Measured detection limit by posterior check (nGy _{air} /s)
0V	4.9	12
Reverse 5V	2.7	5
Reverse 10V	6.9	24
Reverse 20V	14.6	61
Forward 2V	77.1	150
Forward 10V	45.8	150

Personally, I dislike some of the vocabularies used to describe the detector operation in the experimental part. The operation of the photoconductor is called “charge injection” mode. Why is that, are the gold contacts not forming an ohmic contact? In reverse direction the description uses a “suppression” of dark current. However, in the ideal diode equation the current in reverse direction is not called “suppressed” current but rather reverse saturation current.

We appreciate the comment. We have made the best efforts to make our expressions accurate and easily understood by diverse readers while unveiling the fundamental physics behind the terms. The physics behind the term “charge injection mode” and “charge collection mode” were summarized concisely to replace these two terms. The “suppression” of dark current was also replaced by “reverse saturation current”.

Please see changes in Page 13 Line 246-256 in the manuscript.

“According to the I-V behavior, the Pb/MAPbI₃/Au architecture can be considered as photodiode structure while the Au/MAPbI₃/Au architecture is considered as photoconductor structure.

The energy band diagram of a reversely biased Pb/Au device is shown in **Figure S3**, showing that the charge injection from metal electrode into perovskite is negligible due to the energy barrier at the metal-semiconductor interface, which results in the known fact that the photoconductive gain for a reversely biased photodiode is at most 1 because no charge carrier re-circulating could happen due to the negligible charge injection into the semiconductor^{37,38}. For a forward biased Pb/Au device, charge carrier can flow freely from metal electrode into perovskite due to the negligible energy barrier at metal-semiconductor interface (see **Figure S3** for energy band diagram), which results in the photoconductive gain > 1 as charge carrier re-circulating is possible^{37,38}.”

Please see changes in Page 12 Line 242 in the manuscript.

“The Pb/Au devices show a clear current rectifying behavior with a small reverse saturation current (electric field direction under reverse bias is from Pb to Au)

What I find also an unlucky naming is what is described here as “sample size effect”. The sample size could also mean in this manuscript the dimensions of the perovskite specimens.

As an interdisciplinary research, our paper involves both photonics and statistical terminologies. We also recognize the possible confusion by the use of “sample size” in its statistical meaning while it could also mean the number of samples analyzed in radioanalytical chemistry (the original Currie paper), or the physical size of the specimens as reviewer commented. The “sample size” in our paper physically refers to the number of digitized current points when measuring the current signal.

Originally, we used “sample size” as a statistical terminology for a quick assessment and understanding of our method from a statistical point of view. To eliminate confusion and accommodate a diverse readership, we have associated the “sample size” with its direct physical meaning, i.e., “number of data points” or “digitally sampled current data points” as much as possible. The “sample size effect” was also removed.

Please see our changes in Page 6 Line 103-108 in the manuscript.

“...where n stands for the sample size that is a statistical terminology defined as the number of individual samples measured in an experiment. In γ -ray photon counting problems dealing with radioactive decay, the sample size n refers to the number of repetitive time-accumulated counting events the experimenter has made. Each sample counting is lasting for a certain amount of time and have different counts accumulated.”

Please see our changes in Page 7 Line 145 in the manuscript.

“The statistical distribution applicable to photon counting is Poisson-Normal distribution where the sample size n that refers to number of counting performed, is typically 1 or not far larger than 1. Contrarily, the dark current and gross current for an X-ray detector follows Normal distribution, where the sample size n in the current measurement referred as the number of digitized current points is typically very large.”

Please see our changes in Page 8 Line 157 in the manuscript.

“we propose our method based on a statistical model for comparing the means of two normally distributed samples (*i.e.*, physical parameters) of unequal sample size (*i.e.*, number of sample data points), which considers the possibly different standard deviations of the dark current, the gross current, and the large number of digitized current data points”

Thus, in total I find this manuscript in the present form rather not suitable for publication in Nature Communications.

With the revision that has 1) highlighted the breakthrough part of our work; 2) addressed the concern about the broad readership 3) clarified any confusion or misunderstanding of terminologies, and 4.) provided a practical application for determination of X-ray dose rate detection limit, we believe that our paper is suitable for Nature Communications.

Thanks again for the comments.

Reviewer #2 (Remarks to the Author):

In this manuscript, the authors reported a different method to determine the detection limit of X-ray detectors. This is publishable result, but not for Nature Communications, since it fits better in a specialized journal for the reasons listed below.

While we appreciate the comments, we respectively disagree with the notion that our paper is not suited for Nature Communications. Clearly, there is a surge in the number of publications in the high impact journals on the topic of developing perovskite for X-ray detection. **Figure 1** has summarized many of these prior publications, and a few of them are highlighted at below.

1. Wei, W. *et al.* Monolithic integration of hybrid perovskite single crystals with heterogenous substrate for highly sensitive X-ray imaging. *Nat. Photonics* (2017) doi:10.1038/nphoton.2017.43
2. Huang, Y. *et al.* A-site Cation Engineering for Highly Efficient MAPbI₃ Single-Crystal X-ray Detector. *Angewandte Chemie - International Edition* (2019) doi:10.1002/anie.201911281.
3. Pan, W. *et al.* Hot-Pressed CsPbBr₃ Quasi-Monocrystalline Film for Sensitive Direct X-ray Detection. *Adv. Mater.* (2019) doi:10.1002/adma.201904405.
4. Zhuang, R. *et al.* Highly sensitive X-ray detector made of layered perovskite-like (NH₄)₃Bi₂I₉ single crystal with anisotropic response. *Nat. Photonics* (2019) doi:10.1038/s41566-019-0466-7.

5. Zhang, Y. *et al.* Nucleation-controlled growth of superior lead-free perovskite Cs₃Bi₂I₉ single-crystals for high-performance X-ray detection. *Nat. Commun.* (2020) doi:10.1038/s41467-020-16034-w.
6. Pan, W. *et al.* Cs₂AgBiBr₆ single-crystal X-ray detectors with a low detection limit. *Nat. Photonics* (2017) doi:10.1038/s41566-017-0012-4.

Interests for perovskite's X-ray applications has already seen its way into medical X-ray imaging community. Dr. George Zubal, director of the nuclear medicine and computed tomography programs at the National Institute of Biomedical Imaging and Bioengineering (NIBIB), in Bethesda, MD, quoted that "*Interest in perovskite crystals for imaging emerged out of all the recent enthusiasm to get better solar panels,*" in this news published by IEEE Spectrum 56.5 (2019): 10-11, titled "'X-ray detection may be perovskites' killer app: The wonder crystal could yield imagers that are far more sensitive than commercial detectors-[News]." This NIH program funds research into new imaging devices, procedures, and software, including groups looking at perovskite X-ray detection.

With all these rapid developments, there is a lack of clarification in the way X-ray detection limits are determined. Please see our changes in Page 2 Lines 36-42 for current ambiguity.

"Many publications^{12-14,16-19} have only reported the sensitivity of their perovskite X-ray detectors, while lacking the measure of the detection limits (**Figure 1**). Among those that have reported the detection limit (**Figure 1**), some papers^{7,9,20} did not present clearly the method for their detection limit determination, and some²¹⁻²⁵ claimed the use of a method based on the 1975 International Union of Pure and Applied Chemistry (IUPAC) detection limit definition²⁶ without an explicit equation provided. Although some of the work²⁷⁻²⁹ have also adopted a method referenced as derivatives of the IUPAC definitions, there has been no discussions on its statistical validity."

We believe our model and method helps to clarify the current ambiguity and establishes principles for X-ray detector detection limit determination. Please see revised text in Page 7 Line 140 for detailed analysis.

"A simple modification to the IUPAC definition would not result in a correct equation to use for detection limit determination because of the different physics involved. IUPAC definition follows the Currie method that is applicable for γ -ray photon counting where the physics behind is radioactive decay. However, in X-ray detection, the large quantity of X-ray photons emitted from X-ray machine is not a decay phenomenon in nature, nor does the dark current and the gross current of an X-ray detector. The statistical distribution applicable to photon counting is Poisson-Normal distribution where the sample size n that refers to number of counting performed, is typically 1 or not far larger than 1. Contrarily, the dark current and gross current for an X-ray detector follows Normal distribution, where the sample size n in the current measurement referred as the number of digitized current points is typically very large."

Not only do we believe our paper fits well to Nature Communication, we also felt the urgency to publish our paper. As the reviewer #1 commented, our new method is useful as a guideline for a certification laboratory, and we also believe that our method will be broadly adopted and cited by incoming papers in many years presenting X-ray detection limit. For easy adoption, we provide practical procedures to characterize and compare X-ray detection limits at the end of the paper. Please see changes in Page 19 Line 361.

"A practical procedure of both *prior* calculation and *posterior* check is provided below in **Figure**

5 for an easy adoption.

Figure 5. a. practical procedures of theoretical detection limit determination by *prior* calculation **b.** practical procedures of detection limit experimental measurement by *posterior* check. $\sigma_{I_{gross}}$ is the standard deviation of the gross current measured with X-ray. $n_{I_{gross}}$ and $n_{I_{dark}}$ are the number of digitized current points of gross current and dark current, respectively. $\mu_{I_{gross}}$ and $\mu_{I_{dark}}$ are the measured mean value of gross current and dark current, respectively.

The difference of the detection limit is most likely caused by the material quality of contact, rather than the method of characterization.

We absolutely agree that the detection limit is determined by the material quality, surface treatment, as well as the device operation mode and device structure (e.g., reversely vs forward biased photodiode). The detection limits should not be affected by how people measure it. This is exactly why do we need to have a standard method of characterization so that the performance of the perovskite X-ray detector can be compared under the fair ground, which will then help to improve materials quality and device fabrication techniques.

Please see changes in Page 19, Line 361 for the added **Figure 5** for the practical procedure.

The method shown here do not provide direct guidance in X-ray imaging application where the algorithm of imaging collection and process can be very different. What is shown in

literature so far does not represent the real imaging process either, but the method represented here did not improve much.

Thanks for the comment.

Development of perovskite for X-ray detection is still in its infancy where the discussions of sensitivity and lower X-ray detection limit are meaningful, since almost all groups are developing single devices at this early stage as opposed to fabricating multi-million pixelated X-ray imager that is able to acquire a spatially resolved image with one single X-ray scan. It is appropriate to characterize lower detection limits at this early stage since it provides a comparable standard as a feedback to material synthesis in order to advance it to the next phase.

When the development is matured to the stage of pixelated imaging device level, the industrial accepted evaluation standards such as modulation transfer function (MTF), noise power spectrum (NPS) and Detective Quantum Efficiency (DQE) will apply. Therefore, we do agree with reviewer that the method shown here do not provide direct guidance in X-ray imaging application because we are not there yet.

In fact, this paper would fit better in a specialized journal than Nature Communications if we were to focus on MTF, NPS and DQE. Please see our changes in Page 20 Lines 368-372.

“When the development of perovskite X-ray detectors is advanced to the pixelated imager with a single readout of X-ray scan, the system level characterization metrics for spatial resolution and noise performance such as modulation transfer function, noise power spectrum, and detective quantum efficiency may be better suited for their characterizations at that stage.”

The whole content does not have to be related to perovskites.

That’s true. Our paper presents a broader application that goes beyond perovskite. However, the development of perovskite is the focus of the time and represents a broad readership and interests.

The time scale in Figure 4a is not comparable to real X-ray imaging.

Agreed, this is a correct observation! Same as our response in the previous comments, the development of perovskite for X-ray detection hasn’t advanced to the stage of performing real X-ray imaging. The evaluation metric for an X-ray imager will be using MTF for resolution and NPS and DQE for noise performance. The time scale for a real X-ray imaging will be much shorter than the current practice that is still focused on single detection evaluation, with none of them have been able to fabricate pixelated device with ASIC readout. Most of the X-ray imaging picture presented by the state-of-the-art is either scanning detector or the X-ray source to map out current readout in order to produce a 2D picture. The objectives of evaluating perovskite X-ray sensitivity and/or detection limits are to provide feedback for a better materials growth, the time scale of hours or minutes is appropriate for this purpose.

Reviewer #3 (Remarks to the Author):

The authors well studied determination of X-ray detection limit for various perovskite X-ray detectors. The whole study is highly important for this field and the result could help build discipline to quantitatively judge the performance. There are still several issues to be addressed.

Thanks for the comments.

1. The charge collection and injection mode could result in totally different dark current and sensitivity. The prior detection limit seems suitable for charge collection mode. However, for charge injection mode, the photoconductive gain effect will increase the noise value (also known as generation-recombination noise) for signal current. Thereby, it will not correct to use the dark current to calculate the detection limit for collection mode. The detection limit for #2 at reverse/forward bias should be quantitatively compared, not in the level of orders.

We appreciate the comment. We agree that the different noise value of the dark current and the signal current (named as “gross current” in our paper) is a critical issue to be considered for detection limit determination. This is where our method is mathematically different than the currently practiced method in ref [27-29]. Please see our revisions in manuscript for a detailed analysis.

Please see changes at Page 7 Lines 129-137.

“ In the IUPAC definition, the noise value is taken as the standard deviation of the blank signal, σ_B , as it follows the Currie method assuming the standard deviation is the same for blank signal and for gross signal, that is, $\sigma_B = \sigma_Q$. However, practically, the noise current I_{noise} taken as the standard deviation of gross current $\sigma_{I_{gross}}$ will typically be larger than the noise current taken as the standard deviation of dark current $\sigma_{I_{dark}}$ due to the fluctuation of X-ray photon flux, X-ray generated charge carrier generation and recombination. The practiced method²⁷⁻²⁹, shown in **Figure 2b**, considered such difference by simply replacing $\sigma_{I_{dark}}$ in the IUPAC definition with $\sigma_{I_{gross}}$, which lacks statistical validity and results in an improper equation to use.”

Please see changes at Page 8 Line 157.

“we propose our method based on a statistical model for comparing the means of two normally distributed samples (*i.e.*, physical parameters) of unequal sample size (*i.e.*, number of sample data points), which considers the possibly different standard deviations of the dark current, the gross current, and the large number of digitized current data points”

Please see changes at Page 10 Lines 177-179.

“The standard deviation of the gross signal and the blank signal are assumed approximately equal as it is in the Currie formulars, *i.e.*, $\sigma_1^2 = \sigma_2^2$, when the gross signal is small approaching the level of detection limit.”

The *prior* calculation aims at calculating theoretical detection limit of signal (gross) current when signal (gross) current is small enough close to the detection limit. Under such small current, it is a valid assumption that dark current and signal (gross) current have approximately the same noise

value, which is the same way how Currie made assumption on the noise values in his classical paper. Hence, the *prior* calculation method should be applicable to either device operation mode.

To present a systematic quantitative comparison of the sensitivity and detection limit of the #2 device under reverse/forward bias mode, we added a table (**Table 3**) at Page 17, Line 327 to the manuscript.

“A quantitative comparison of the sensitivity and detection limit of #2 Pb/Au device under reverse vs forward bias mode was presented in **Table 3**”

Table 3. #2 Pb/Au device detection limit and sensitivity under reverse vs forward bias mode

	Reverse 2V	Forward 2V	Reverse 10V	Forward 10V
Detection limit (nGy _{air} /s)	2.4	77.1	6.9	45.8
Sensitivity (μC/Gy _{air} /cm ²)	1.1×10 ⁴	3.5×10 ⁵	2.0×10 ⁴	1.5×10 ⁶

2. Moreover, according to Figure 4a, the dark current is drifting in the measurement period. Then how to determine the exact dark current value. Also, will the ionic drifting in perovskite influence the detection limit. For perovskite detectors, the ionic drifting current is sometimes increasing and sometimes decreasing, which will have totally different impact on the dark current value. But the ionic drifting is not involved in their equation.

Thanks for the comment. We agree that the ionic drifting which originates from materials property does affect the detection limits, just like other physical factors, e.g., internal material defects or external temperature effects. However, they should not be changing the way or method of how detection limit is characterized. The physical factors influencing the detection limit are included implicitly in the measured dark current values, so the physical factors will not appear explicitly in the pure statistical equation. Please see our revision at Page 4 Lines 68-72.

“We conclude that the device architecture and operation mode have a significant influence on the value of sensitivity and detection limit to be measured, but our proposed methodology for the determination of detection limit is independent of the material’s properties and device’s operation, which could be used as an evaluation standard for materials quality and detector performance comparison.”

The dark current drifting in **Figure 4a** will influence the dark current measurement and hence eventually the detection limit of the device. To mitigate the effect of dark current drifting on the accuracy of calculated detection limit, we purposely use the dark current data after a long time of X-ray turn on (e.g., more than 100s in **Figure 4a**) where the drifting becomes minor. Please see our revision at Page 15 Lines 284-285.

“The dark current standard deviation, *i.e.*, σ_2 , is ~ 0.376 pA calculated using the dark current that has a minor drifting after a long time of biasing,”

3. The authors should give some comments on what parameters could best express the performance of perovskite detectors. High sensitivity or low detection limit?

We appreciate the constructive comments. As suggested, to best express the performance of perovskite detectors, we made revisions by adding a figure (**Figure 4d**) to show qualitatively the device operation condition's influence, e.g., reversely vs forward biased photodiode, on detection limit and sensitivity. We also added a table (**Table 3**) to show quantitatively such influence.

Please see Page 18 Line 332-340.

“To present a comprehensive picture of device architecture and operation mode’s influence on the sensitivity and the detection limit, we qualitatively illustrate the current value as a function of applied bias voltage for a photodiode working under forward vs reverse bias mode in **Figure 4d**. Although the photodiode under forward bias mode has a larger sensitivity (slope of the linear fitting) than that under reverse bias mode, the forward bias mode also has a larger dark current. The X-ray dose rate cannot be detected infinitesimally. Instead, the X-ray dose rate detection limit \dot{D}_{limit} is depending on the photocurrent detection limit I_{limit} and the device sensitivity s , and I_{limit} is further determined by the dark current. The specific quantitative relationship between I_{limit} and the dark current is established by the statistical model in this paper.

Figure 4d. qualitative comparison of sensitivity (slope of the linear fitting) and dark current for photodiode working under forward vs reverse bias mode.

Please see Page 17 Line 328.

A quantitative comparison of the sensitivity and detection limit of #2 Pb/Au device under reverse vs forward bias mode was presented in **Table 3**.

Table 3. #2 Pb/Au device detection limit and sensitivity under reverse vs forward bias mode

	Reverse 2V	Forward 2V	Reverse 10V	Forward 10V
Detection limit (nGy _{air} /s)	2.4	77.1	6.9	45.8
Sensitivity (μC/Gy _{air} /cm ²)	1.1×10 ⁴	3.5×10 ⁵	2.0×10 ⁴	1.5×10 ⁶

4. Figure 1a summarizes the sensitivity and detection limit for reported results. Are these results following the prior detection limit value? The author should give a check, which is important for the general applicability of this paper.

We appreciate the constructive comments. As suggested, we made revision to the manuscript to show the method of detection limit determination for each paper reviewed in **Figure 1**. These papers either did not provide an equation or are based on an improper equation, which shows the current lack or ambiguity of methodology for X-ray detection limit determination. Please see changes in Page 2 Lines 36-42.

“Many publications^{12-14,16-19} have only reported the sensitivity of their perovskite X-ray detectors, while lacking the measure of the detection limits (**Figure 1**). Among those that have reported the detection limit (**Figure 1**), some papers^{7,9,20} did not present clearly the method for their detection limit determination, and some²¹⁻²⁵ claimed the use of a method based on the 1975 International Union of Pure and Applied Chemistry (IUPAC) detection limit definition²⁶ without an explicit equation provided. Although some of the work²⁷⁻²⁹ have also adopted a method referenced as derivatives of the IUPAC definitions, there has been no discussions on its statistical validity.”

REVIEWERS' COMMENTS

Reviewer #1 (Remarks to the Author):

I would like to thank the authors for their efforts to make this paper easier to read and also to change some of the vocabularies. I start to like the idea to get a recipe which can be applied to obtain reasonable numbers for the detection limit of an X-ray detector (perovskite or not is not important), which I consider to one of the most important figure-of-merit of a detector (possibly also for a detector operating in the visible). What I still do not like are some of the vocabularies to be used. In my mind the method which is called here theoretical detection limit determination, is not a theoretical one, because it is based on experimental results. I find also the word "prior" not suitable. Equally un-suitable I find the name "posterior check". These are two methods, providing systematically different results, which I would rather call the "dark current standard deviation method" and the "dark current / gross current" method. Both methods provide a systematic procedure to be done, which are possibly useful, as long it is stated which method is applied. Possibly the methods are not more useful or less work than evaluation of the signal to noise ratio. Why the photocurrent plus dark current (= current under radiation) has to be named now gross current is still not clear to me. I think that doing interdisciplinary research also means to present the results in a manner, that they can be understood within the field for which the methods should be applied: this field is defined in the title as "perovskite X-ray detectors" and not statistical analysis. Thus, the already introduced nomenclature of this research area should be used. The obtained values for the detection limit are anyway good, so that I changed my mind – this paper can be published in Nature Communications, however, after some further changes, taking into account the discussion above.

Reviewer #2 (Remarks to the Author):

Thanks for the author's explanation. I want to clarify that this reviewer does think X-ray detection using perovskite is important. However, this paper only address the question on how the different lab should measure their detector to tell the lowest dose rate of single pixel detector. This info is not helpful in real X-ray imaging detector development, as concurred by the authors. I would think it is a good commentary paper, rather a research paper to Nature Communications.

Reviewer #3 (Remarks to the Author):

The authors have carefully addressed the questions. It is acceptable for Nature Communications.

Response to Reviewer Comments:

Dear Editor and Reviewers,

We sincerely thank you for reviewing our manuscript and providing your valuable suggestions. We believe that we have made every effort to address your comments.

In the following paragraphs, please find our responses to each comment including the changes we have made in the original manuscript. The reviewers' comments are in 'bold' font, while our responses are in regular font. Our revisions in the manuscript are highlighted in green.

REVIEWERS' COMMENTS

Reviewer #1 (Remarks to the Author):

I would like to thank the authors for their efforts to make this paper easier to read and also to change some of the vocabularies. I start to like the idea to get a recipe which can be applied to obtain reasonable numbers for the detection limit of an X-ray detector (perovskite or not is not important), which I consider to one of the most important figure-of-merit of a detector (possibly also for a detector operating in the visible). What I still do not like are some of the vocabularies to be used. In my mind the method which is called here theoretical detection limit determination, is not a theoretical one, because it is based on experimental results. I find also the word "prior" not suitable. Equally un-suitable I find the name "posterior check". These are two methods, providing systematically different results, which I would rather call the "dark current standard deviation method" and the "dark current / gross current" method. Both methods provide a systematic procedure to be done, which are possibly useful, as long it is stated which method is applied. Possibly the methods are not more useful or less work than evaluation of the signal to noise ratio. Why the photocurrent plus dark current (= current under radiation) has to be named now gross current is still not clear to me. I think that doing interdisciplinary research also means to present the results in a manner, that they can be understood within the field for which the methods should be applied: this field is defined in the title as "perovskite X-ray detectors" and not statistical analysis. Thus, the already introduced nomenclature of this research area should be used. The obtained values for the detection limit are anyway good, so that I changed my mind – this paper can be published in Nature Communications, however, after some further changes, taking into account the discussion above.

We sincerely appreciate the reviewer's comments. We agree with reviewer's comments about the choice of vocabulary. Since our targeted audience are sensor and photonics community, the use of statistical terminologies should be reduced to the minimum and the language should be easily understood by the primary audience.

We have made revisions in the manuscript where "gross current" is replaced by "photocurrent under X-ray irradiation I_{X-ray} ". The "prior calculation" is changed to "dark current (I_{dark} & s) method". The "posterior check" is changed to "X-ray photocurrent (I_{dark} & I_{X-ray}) method", which is more straightforward for the people in the field to

understand it. The naming of these methods is trying to be simple and the detailed explanations are given in the text.

We mentioned “*a priori*” one time in the text, only to indicate the statistical nature of the proposed methods. We did the same for “*posterior*”, also only mentioned it once in the text, just trying to lay a proper statistics context.

Please see the changes made in Page 1 Lines 14-17 in the manuscript.

“The detection limit can be calculated through the measurement of dark current and sensitivity with an easy-to-follow practice. Alternatively, the detection limit may also be obtained by the measurement of dark current and photocurrent when repeatedly lowering the X-ray dose rate.”

Please see other changes in Page 3 Lines 56-61.

“Specifically, detection limit of X-ray dose rate \dot{D}_{limit} can be obtained by calculation from a measurement of the device’s dark current I_{dark} and the sensitivity s as a calibration factor, which is, for simplicity, named as dark current (I_{dark} & s) method thereafter. Alternatively, \dot{D}_{limit} may also be obtained by the repetitive measurements of I_{dark} and detector photocurrent under X-ray irradiation I_{X-ray} when repeatedly lowering the X-ray dose rate, which is named as X-ray photocurrent (I_{dark} & I_{X-ray}) method.”

Also in Page 7 Lines 128-131.

“In X-ray detector testing, the blank signal is the dark current I_{dark} and the gross signal is the photocurrent under X-ray irradiation I_{X-ray} . The net current I_{net} is the difference between the photocurrent under X-ray I_{X-ray} and the dark current I_{dark} , *i.e.*, $I_{net} = |I_{X-ray} - I_{dark}|$.”

Also in Page 11 Lines 205-207.

“It is worth to note that the calculation so far only yields the detection limit of net current I_{limit} . The detection limit of X-ray dose rate \dot{D}_{limit} requires I_{limit} and sensitivity s as calibration factor, which will be discussed in the later section”

Also in Page 12 Lines 221-226.

“Following the principles above, we can successively lower the X-ray dose rate used to generate the I_{X-ray} until I_{X-ray} cannot be detected against the dark current. The lowest X-ray dose rate that generates the smallest detectable I_{X-ray} is taken as the dose rate detection limit \dot{D}_{limit} . Although this method needs the measurement of I_{dark} and I_{X-ray} , we simplify the name by calling it the X-ray photocurrent method (I_{dark} & I_{X-ray} method).”

Also in Figure 5.

Fig. 5. Practical procedure for detection limit determination. a. the dark current I_{dark} & s method. N is the number of digitized current points of dark current and I_i is the i^{th} point. **b.** the X-ray photocurrent I_{dark} & I_{X-ray} method. $\sigma_{I_{X-ray}}$ is the standard deviation of the photocurrent under X-ray irradiation I_{X-ray} . $n_{I_{X-ray}}$ and $n_{I_{dark}}$ are the number of digitized current points of I_{X-ray} and I_{dark} , respectively. $\mu_{I_{X-ray}}$ and $\mu_{I_{dark}}$ are the measured mean value of I_{X-ray} and I_{dark} , respectively.